# Advancements in Circulating Tumor Cell Research: Bridging Biology and Clinical Applications

**DOI:** 10.3390/cancers16061213

**Published:** 2024-03-20

**Authors:** Philip Salu, Katie M. Reindl

**Affiliations:** 1Cellular and Molecular Biology, North Dakota State University, Fargo, ND 58105, USA; philip.salu@ndsu.edu; 2Department of Biological Sciences, North Dakota State University, Fargo, ND 58105, USA

**Keywords:** circulating tumor cells, liquid biopsy, cancer research, metastasis, personalized medicine, single-cell sequencing, biomarkers

## Abstract

**Simple Summary:**

Cancer remains one of the leading causes of death worldwide due to widespread metastasis. Circulating tumor cells (CTCs) were first detected in 1886 and have been thought of as seeds of metastasis. However, emerging evidence has shown that CTCs can be used for early disease detection and as predictors of disease prognosis and treatment response. This review discusses recent findings regarding the detection and characterization of CTCs. Integrating these findings into routine clinical workflows promises to revolutionize the diagnosis and treatment of cancer.

**Abstract:**

Circulating tumor cells (CTCs) are cells released from the primary and metastatic tumor and intravasate into the blood or lymphatic vessels, where they are transported to distant sites and act as seeds that initiate cancer metastases or the development of further lesions. Recent advances in CTC research have shown their relevance as prognostic markers for early and metastatic disease detection, predictive biomarkers for relapse, and response to medical intervention or therapy. The rapidly evolving landscape of CTC biology has opened new avenues for understanding cancer progression, metastasis, and treatment response. Additionally, translating these findings into clinical applications holds promise for improving cancer diagnostics, prognosis, and personalized therapeutic strategies. This review discusses the significance of CTCs in cancer research and their associated challenges. We explore recent developments in the detection and characterization of CTCs and their implications in cancer research and clinical practice.

## 1. Introduction

In the United States (US), cancer remains the second leading cause of death, affecting individuals of all ages and ethnic groups [1]. The estimated number of all cancer-related deaths in the US is over 609,000, which is about 18% of all US deaths [1]. Cancer mortality is due to poor treatment response, metastasis, and the lack of early detection methods [2,3,4]. Liquid biopsies are gaining popularity as noninvasive techniques for early disease detection [5,6]. These liquid biopsies contain circulating tumor cells (CTCs) that are released into the blood by a tumor and travel through the bloodstream to other areas of the body to form metastatic niches (Figure 1) [7].

CTCs are proving to be essential biomarkers that can be used for early disease detection, monitoring metastasis and cancer progression, and determining genetic heterogeneity in tumors [8,9,10]. Compared to traditional tumor biopsies, obtaining CTCs from liquid biopsies is non-invasive and real-time [11]. This can be more acceptable to patients, paving the way for more personalized cancer medicine.

However, several challenges currently affect CTC research. One of the main issues is that they are rare and heterogeneous [12,13]. CTCs are present in very small numbers, about 1–100 per 10^9^ white blood cells [14]. These scarcely populated cells are characterized by varying morphology, phenotypical characteristics, and genetic makeup, making isolation and characterization difficult [15,16]. Moreover, standardizing isolation techniques and establishing their clinical relevance presents ongoing challenges [17,18]. Ethical considerations and the need for more extensive clinical trials also pose significant obstacles to advancing CTC research [19]. Available techniques for detecting CTCs, like flow cytometry-based and microscopy-based detection methods, need refinements to improve accuracy and sensitivity [20].

Current approaches to the isolation and analysis of CTC biology involve using microfluidic devices, single-cell sequencing, and the integration of multi-omics data to determine the molecular and functional heterogeneity of CTCs [21,22,23]. Machine learning tools trained to interpret complex CTC data are also being developed to aid in identifying biomarkers and potential therapeutic targets [24,25]. These methods hold promise for improved detection, monitoring, and uncovering treatment-resistant or metastatic CTC subpopulations. This review discusses the significance of CTCs in cancer research, current challenges, and new approaches to characterizing CTC tumor cell biology. Additionally, we preview the latest findings in CTC research and their clinical implementations in cancer.

## 2. Significance of CTCs in Cancer and Clinical Implications

Circulating tumor cells hold immense significance in cancer research and clinical applications due to their potential to offer valuable insights into various aspects of cancer biology and patient care. Recent discoveries in CTC research have significantly advanced our understanding of their roles in cancer progression, metastasis, treatment response, and personalized medicine. These new developments hold significant clinical implications, potentially revolutionizing various aspects of cancer. Below, we discuss recent insights into early cancer detection, metastatic potential, prognosis, treatment monitoring, and personalized therapy based on CTC characterization and their potential clinical impacts.

### 2.1. Early Cancer Detection

Early detection of cancer is crucial for effective treatment and disease management to reduce cancer-related deaths and improve patient outcomes. CTCs offer the potential for liquid biopsy-based screening, allowing the identification of cancers at an earlier, more treatable stage [26]. CTCs were first detected in cancer in the eighteenth century [27]. However, the process of local invasion and extravasation of CTCs has recently been shown to occur within a short span of time [28], suggesting that CTCs can be detected in the bloodstream at early stages of cancer, potentially allowing for early diagnosis before tumors become clinically evident [29,30]. A study involving 667 participants, of whom 235 were healthy individuals and 432 were patients with either colorectal cancer (CRC) or adenomas, showed that CTCs can be used to distinguish healthy individuals from patients with CRC or adenomas with a detection sensitivity of up to 95.2%, even before colonoscopy [31]. Another study evaluating patients with chronic obstructive pulmonary disease (COPD) without clinically diagnosed lung cancer revealed that five out of 168 patients had detectable CTCs. Notably, all five individuals with detectable CTCs developed lung nodules within 1–4 years [32]. A third study involving patients with suspected prostate cancer based on high serum prostate-specific antigen (PSA) levels also revealed a high positive CTC-predicted biopsy outcome and prostate cancer aggressiveness [33]. Thus, the importance of using CTCs to predict disease occurrence cannot be underestimated. Additionally, capturing and analyzing CTCs using liquid biopsy techniques offer a non-invasive alternative to traditional tissue biopsies, providing a real-time snapshot of the tumor’s genetic and phenotypic characteristics.

### 2.2. Prognostic Indicators

The presence of CTCs in the bloodstream provides valuable prognostic information, aiding in predicting disease outcomes and guiding treatment decisions [34,35,36]. Studies have shown that higher CTC counts and the presence of CTC clusters are associated with poorer outcomes across various cancer types [37,38]. In clinical settings, CTCs are commonly used as a liquid biopsy method to screen for tumors, monitor disease, and predict patients’ prognoses [39]. A CTC count cutoff value of ≥3 CTCs/7.5 mL blood is considered unfavorable for metastatic colorectal cancer [40]. In a study involving hormone receptor-positive (HR+) metastatic breast cancer patients, the overall survival (OS) and progression-free survival (PFS) of patients with CTC counts ≥ 5 per 7.5 mL blood after treatment was significantly worse than those of patients with <5 CTCs [41]. Also, CTC counts ≥ 10 per 5 mL blood in small-cell lung cancer (SCLC) patients were closely associated with advanced stage (high lymph node metastasis and distant metastasis), indicating a more unfavorable prognosis [42]. These recent investigations have refined our understanding of CTCs as robust prognostic indicators, guiding clinicians in predicting disease outcomes and tailoring treatment strategies. Multiple detection methods have been developed to comprehensively characterize CTCs, providing valuable insights into the likely cause of disease [39,43]. CTC enumeration is increasingly being incorporated into clinical practice as a predictive tool, and CTC counts are being used to stratify patients, helping tailor treatment plans based on individual prognoses [44].

### 2.3. Tumor Heterogeneity

Tumor cells obtained from liquid biopsies exhibit genetic and phenotypic diversity, reflecting heterogeneity within the primary tumor [13,45]. Numerous studies have discovered the presence of different CTC subpopulations in cancers [13,46,47,48]. For instance, Freed and colleagues identified two CTC subpopulations expressing epithelial cell adhesion molecule, EpCAM (CTC^EpCAM^), or fibroblast activation protein-alpha, FAPα (CTC^FAPα^), in pancreatic ductal adenocarcinoma (PDAC) patients. Using the ratios of CTC^FAPα^ to CTC^EpCAM^, they were able to stratify patients as responders versus non-responders to niraparib treatment [46]. In a non-small cell lung cancer (NSCLC) study, the authors were able to detect different gene signatures related to therapy resistance (MET and HER3) and the initiation of metastasis (ALHD1) using CTCs [49]. The spectrum of cancer heterogeneity has long been under scrutiny as a factor allowing the tumor to adapt to different microenvironmental stressors with different subpopulations that may evolve to increase disease aggressiveness and decrease therapeutic response [50]. The study of CTCs, therefore, allows researchers to understand and monitor tumor heterogeneity, providing crucial information for devising effective and personalized treatment strategies.

### 2.4. Metastasis and Disease Progression

Metastasis accounts for more than 90% of cancer-related mortalities [51,52,53]. This is because treatment options available for patients with metastatic solid tumors are rarely curative [54,55]. Metastatic events in cancer involve the spread of tumor cells from the primary to secondary sites, following a phenotypic transition from epithelial to mesenchymal phenotype, cancer cell invasion into circulation, dormancy, and colonization at distant sites [56,57]. CTCs can be released by aggressive and metastatic tumors into circulation, where they extravasate to other remote sites for continuous colonization, leading to the growth of further lesions [58]. Nonetheless, research suggests that CTCs can be shed from primary tumors at early stages of cancer, challenging the traditional view that metastasis occurs primarily in advanced disease [6]. This early dissemination contributes to the understanding of metastatic potential and the dynamics of cancer progression [6,59]. Following the injection of pancreatic cancer cells expressing fluorescent proteins into the earlobes of mice to form solid tumors, the presence of CTCs with fluorescent proteins was detected in the bloodstream of the mice at different stages of development [60]. Using quantum dots, the researchers identified cells with high metastatic potential; those expressing clusters of differentiation, CD24+ and CD133+ [60]. The development of metastatic niches often signifies a significant progression in the tumor stage, and studies have linked CTC characteristics, such as the presence of CTC clusters, to increased metastatic potential, aiding in the identification of patients at higher risk [61,62,63]. In metastatic breast cancer, a CTC count ≥ 5 in 7.5 mL of blood correlates with worse overall survival and progression-free survival [40]. Indeed, a CTC count of less than five in patients with stage IV breast cancer is used to classify the tumor as stage IV indolent, whereas a CTC count greater than five classifies the tumor as stage IV aggressive [40]. CTC clusters exhibit stemness characteristics and can evade the immune system by recruiting immunosuppressive cells [64]. This attribute helps prevent CTCs from being attacked by antitumor immune cells like natural killer (NK) cells, thereby increasing their metastatic potential [64,65,66]. In circulation, CTCs are protected from shear forces and shielded from immune detection through interactions with platelets [67]. Other blood cells, like macrophages, are able to interact directly with CTCs to protect them from being phagocytosed [67]. Szczerba et al. also identified and associated CTC–neutrophil interactions with cell cycle progression, leading to more efficient metastasis formation [68]. Using a mass spectrometry-based untargeted metabolomics approach, human colorectal cancer CTC-derived cells were shown to have low or high metastatic potential based on metabolic features [69]. A combination of metabolites, such as glutamic acid, malic acid, lactic acid, and aspartic acid, along with higher CTC counts, positively predicted the metastatic risk in patients, providing evidence of the influence of metabolic phenotype on the metastatic potential of cells [69]. Recently, CTCs were shown to have unexpectedly high levels of OXPHOS compared to glycolytic signatures [70]. This metabolic reprogramming was the opposite of the common “Warburg Effect” seen in metastatic cancer cells, suggesting an additional layer of regulative complexities in cancer metastasis [70]. Metabolic reprogramming of CTCs enables them to survive harsh conditions in the bloodstream and enhances their ability to establish a favorable microenvironment for metastasis, known as metastatic niches [71]. The molecular mechanisms by which CTCs influence the pre-metastatic niche (PMN), prepare distant organs for colonization, and contribute to the metastatic process are yet to be fully understood. However, CTCs acquire enhanced migratory and invasive capabilities when undergoing epithelial–mesenchymal transition (EMT) [72]. Recent discoveries have elucidated the dynamic nature of EMT in CTCs, identifying hybrid epithelial/mesenchymal states that may contribute to metastatic progression [38]. For instance, the presence of both epithelial and mesenchymal markers indicates the occurrence of intrahepatic metastasis, while the presence of mesenchymal phenotypes prompts the development of extrahepatic metastasis [38]. Expression of mesenchymal markers like N-cadherin, vimentin, snail, and slug have also been detected in CTCs, highlighting their use in metastasis prediction [73,74]. Using CTC-based information to assess the risk of metastasis will help implement more aggressive treatment strategies for patients with a higher likelihood of developing metastatic disease.

### 2.5. Treatment Response Monitoring

Changes in CTC counts and characteristics during treatment can indicate treatment response or resistance [41]. Real-time monitoring of CTC dynamics will allow for timely adjustments to therapeutic strategies. To monitor patient response to therapy, longitudinal tissue biopsies must be collected. However, this procedure is often invasive and complicated by the location and size of tumors [75]. Therefore, CTCs can serve as dynamic biomarkers for monitoring treatment response and assessing the efficacy of therapies. Indeed, CTCs from patients with either chemosensitive or chemorefractory SCLC were tumorigenic in immune-compromised mice, and their resultant CTC-derived explants mirrored the donor patients’ response to platinum and etoposide chemotherapy [76]. Molecular alterations that affect a tumor cell’s response to specific drugs can also be determined through molecular profiling of CTCs. For example, the expression level of cyclin D1 (CCND1) was shown to be significantly reduced in patients with head and neck squamous cell carcinoma (HNSCC) following nivolumab treatment [77]. In contrast, the expression of NANOG increased significantly [77]. Changes in CTC count or specific genetic alterations can, thus, indicate whether a treatment effectively targets the tumor, enabling timely adjustments to the therapeutic approach.

### 2.6. Minimal Residual Disease Monitoring

CTCs can be used to monitor minimal residual disease (MRD) after surgery or other treatments, helping identify potential disease recurrence early. Li and colleagues used preoperative CTC concentration to predict postoperative recurrence or metastasis in patients with NSCLC [78]. Detection of CTCs pre- and post-adjuvant chemotherapy can also serve as prognostic indicators of early relapse [34]. These indicators allow for investigations into novel therapeutic approaches to curb relapse and drug resistance. CTC-based MRD monitoring has also been explored as a valuable tool in post-treatment surveillance, offering insights into the effectiveness of interventions and informing decisions regarding additional therapies [79]. CTCs can enter a state of dormancy, temporarily evading the immune system and therapeutic interventions [80,81]. Recent studies have shed light on the mechanisms governing CTC dormancy and the factors that lead to their reactivation, contributing to a deeper understanding of cancer relapse [82,83,84]. Molecular characterization of CTCs can also help identify and analyze drug-tolerant clones within the tumor microenvironment (TME) [85].

### 2.7. Personalized Medicine

Precision oncology aims to treat patients according to the molecular characteristics of their tumors, adjusting these treatment strategies as the tumors evolve [86]. Molecular characterization of CTCs contributes to identifying actionable mutations and potential therapeutic targets. Whole exome sequencing (WES) of patient CTC samples has been shown to reflect the genomic characteristics of their corresponding solid tumors more accurately [87,88,89]. This genomic information can be crucial in developing personalized treatment plans. Functional mutations in driver genes like EGFR, KRAS, and TP53 can be determined quickly and used to stratify patients for therapy [89]. Mutational characterization of CTCs can also help physicians decipher if and when a patient is developing resistance to a particular chemotherapeutic agent, thereby allowing for proactive treatment decisions [90]. In this regard, CTC-based liquid biopsies facilitate the development of personalized treatment plans, allowing for the selection of targeted therapies based on the specific genetic profile of the tumor. Furthermore, CTC-based liquid biopsies are increasingly integrated into clinical trials and treatment decision-making.

### 2.8. Clinical Trials and Drug Development

CTCs offer a tool for assessing drug response in preclinical and clinical trials, aiding in developing novel cancer therapeutics. Current CTC-based clinical trials aim to use CTC positivity and dynamics to predict clinical outcomes and therapy responses in patients [19]. Following the STIC CTC randomized clinical trial (NCT0170605), researchers determined that CTC count was a reliable biomarker for choosing between chemotherapy and a single-agent endocrine therapy as a first-line treatment in hormone receptor-positive ERBB2-negative metastatic breast cancer [91]. Data from clinical trials can also be used to model the prognostic impact of a tumor and the design of specific targeting agents [92]. Therefore, CTC-based studies provide a platform for evaluating the effectiveness of new treatments and understanding drug resistance mechanisms.

In summary, CTCs have a significant role in cancer research and clinical applications. They act as dynamic biomarkers, providing insights into the complex nature of cancer. With technological advancements, CTC-based approaches can be integrated into regular clinical practice. This can potentially revolutionize cancer diagnosis, treatment, and patient outcomes.

## 3. Challenges of Using CTCs in Research and Clinical Applications

Studying circulating tumor cells presents several challenges, primarily stemming from their rarity in the bloodstream and the technical complexities associated with their isolation and characterization. Despite all the limitations associated with CTCs, rapidly evolving technology is paving the way for cutting-edge detection and characterization methods. Table 1 outlines some key challenges related to the use of CTCs.

## 4. Circulating Tumor Cell Detection Strategies—Pros and Cons

The study of CTCs presents an opportunity to gain valuable insights in advancing cancer research and improving patient care. However, addressing the challenges with CTC research starts with effectively detecting CTCs in liquid biopsies. As such, there is a need to refine existing techniques and develop standardized protocols while making the process cost-effective and accessible to patients [110]. Admittedly, the methodologies used to detect or isolate CTCs have evolved significantly, becoming progressively sophisticated and sensitive [111,112,113,114]. Current CTC detecting methods can be categorized into label-dependent (affinity-based) and label-independent [12]. The latter involves traditional methods that separate cells based on size, deformability, and other biophysical properties [115,116]. These methods range from density gradient centrifugation, inertia focusing, filtration, and dielectrophoresis [117,118]. The challenge is that these methods lack the sensitivity and specificity needed to isolate CTCs efficiently [101].

Label-dependent methods detect cells based on the expression of specific markers like EpCAM, vimentin, and N-cadherin (positive selection), or the lack thereof of markers like CD45 (negative selection) and the binding affinity to specific antigens, RNA, and DNA sequences [101,119,120]. Such methods include EpCAM enrichment, immunomagnetic separation, and microfluidic devices [121,122,123]. By far, the cell surface adhesion molecule, EpCAM, is the most common marker used for CTC enrichment (increasing CTC concentration for their subsequent detection) [124]. Recent studies have focused on improving CTC enrichment through highly engineered microfluidic devices and selective CTC markers [112,124]. To date, label-dependent methods offer improved sensitivity and specificity, enabling the capture of rare CTCs while minimizing contamination from normal blood cells [125,126,127]. Nonetheless, the lack of universal markers for detecting heterogeneous CTCs coupled with the unavailability of validated and standardized approaches means that most of these approaches need to be validated and standardized in the pre-analytical, analytical, and post-analytical phases [128]. Table 2 displays some methods and their principles for detecting CTCs and their advantages and disadvantages.

Commercialized single-cell isolation strategies such as fluorescence-activated cell sorting and droplet-based systems isolate single cells randomly, making isolating rare single cells difficult [16]. The Food and Drugs Authority (FDA) has approved CELLSEARCH for the enumeration of CTCs of epithelial origin (EpCAM+, CD45−, and cytokeratins 8, 18+, and/or 19+) in whole blood [138]. Other systems such as DEPArray [139], RareCytes CyterFinder [140], ALS Collector [141], and VyCap Puncher [142] can isolate rare single cells but lack the capacity for single-cell enrichment. A study by Yoshino and colleagues showed that target cells could be encapsulated by confocal laser-scanning microscopy following the introduction of a photopolymerized hydrogel, polyethylene glycol diacrylate (PEGDA), into cell entrapped on a microcavity array (MCA) [143]. In another study utilizing the MCA/gel-based cell manipulation (GCM), a size-selective CTC enrichment and a surface-independent CTC detection method (based on CellTracker Green-positive and CD45-negative) showed increased recovery and effective detection of gastric cancer CTCs [144]. This GCM-based single-cell manipulation method is advantageous for extremely low-volume reactions due to the minimal water composition of hydrogel-encapsulated single cells [144]. These efforts are necessary to decrease cell loss during sample handling, decrease white blood cell contamination, and increase the recovery of CTCs in cancer.

## 5. Advancements in Characterizing CTC Biology and Clinical Implications

The study of CTCs is inherently challenging due to their rarity in the bloodstream. To overcome this challenge and make the most use of CTCs as valuable biomarkers, advanced technologies are essential for their efficient capture and precise analysis. Herein, we discuss recent progress that has been made in characterizing CTC biology.

### 5.1. Polymerase Chain Reaction (PCR)

Liquid biopsies contain CTCs and other tumor-derived materials shed into the bloodstream, such as circulating tumor nucleic acids, including DNA (ctDNA), RNA (ctRNA), exosomes, and other microvesicles [145,146,147]. ctDNA, derived from apoptotic or necrotic tumor cells and CTCs, can be used to predict changes to therapeutic response [148]. The FDA approved the Cobas EGFR mutational test for the management of therapy in lung cancer [149]. This was a result of the detection and mutational testing (Cobas EGFR mutation test) of ctDNA from NSCLC patients, which showed the importance of ctDNA as a biomarker in cancer diagnosis and prognosis [150,151]. Different types of PCR methods have been developed with the aim of detecting and characterizing CTCs and ctDNA [152,153,154,155]. These methods continue to be improved to increase sensitivity and allow for accurate and quantitative analyses of CTCs [111]. The development of droplet digital PCR (ddPCR) has allowed for the partitioning of a sample into many “droplets,” enabling the amplification of rare mutations [153,156]. The ddPCR method offers several advantages over conventional quantitative and real-time PCR [157,158,159,160]. It has improved sensitivity [158], is less susceptible to PCR inhibitors [157], and allows for absolute quantification without the need for an external calibration curve [160]. CTCs can also be quantified independently of nucleic acid extraction using real-time quantitative PCR (Q-PCR) [155]. In the article by Mei and colleagues, CTCs were isolated using tag-DNA-modified CK19 antibody and magnetic beads conjugated with EpCAM antibody. They reported a detection rate of 92.3% in the clinical tumor blood samples, and by quantifying the tag-DNA that had immobilized on the tumor cells, they showed a correlation between CTC counts and tumor stage/status [155]. As PCR methods improve, single-cell and next-generation sequencing can be integrated to enhance CTC characterization and their roles in cancer.

### 5.2. Single-Cell Analysis

Genomic and phenotypic characteristics of cells differ within a patient’s tumor (intratumor heterogeneity) and between tumors of the same type (intertumor heterogeneity) [161]. These heterogeneous differences contribute to treatment failure and disease recurrence, as there is no one-size-fits-all treatment approach [85]. Evidence has shown that CTCs detected in patients represent an array of cell states with varying genetic and phenotypic profiles [97,162]. Therefore, advanced technologies like single-cell RNA sequencing (scRNA-seq), single-cell protein analysis, and imaging techniques are needed [163]. These techniques allow for detailed molecular and phenotypic characterization with the potential to reveal deep insights into intratumor heterogeneity and cellular plasticity and help identify pathways activated in different cell states, details often masked by profiling pooled cells [163]. Traditional methods of CTC analyses have targeted the use of epithelial biomarkers, whose expression may not be universal across all tumors [120]. Recent improvements in single-cell analysis have focused on enhancing sensitivity, reducing technical variabilities, increasing throughput, and upgrading data analysis tools [164,165,166]. Upgrades in single-cell analysis packages, such as Seurat and Uniform Manifold Approximation and Projection (UMAP), have enabled better clustering analysis and classification of CTCs into multiple subtypes [167,168]. Single-cell sequencing analysis of CTCs from SCLC patients showed a global increase in intratumoral heterogeneity (ITH), including heterogeneous expression of potential therapeutic resistance pathways, like EMT, between different cellular subpopulations following treatment resistance [48]. Indeed, serial profiling of the CTCs post-relapse confirmed increased ITH, suggesting that treatment resistance was characterized by coexisting subpopulations of cells with heterogeneous gene expression, leading to multiple concurrent resistance mechanisms [48]. This finding effectively highlights the need for clinical efforts targeted at developing combination therapies for treatment-naïve SCLC tumors in order to increase initial response and counteract the emergence of ITH and diverse resistance mechanisms [48]. Another analysis of prostate CTCs revealed heterogeneity in signaling pathways like non-canonical Wnt signaling that could contribute to treatment failure or therapeutic resistance (antiandrogen resistance) [169]. By integrating single-cell and batch RNA sequencing, Zhang et al. found gene markers associated with CD8^+^ T cell infiltration (LGR5, CCR7, STC2, DEFB1, TYK2, SCARB1, AICDA, and ULBP2) in HNSCC, which can serve as predictors of disease prognosis and clinical treatment indicators [170]. These findings may change the way cancer patients are staged and risk-stratified and provide new prognostic approaches for the identification of targeted therapy and resistance mechanisms.

### 5.3. Next-Generation Sequencing (NGS)

High-throughput sequencing of CTC-derived DNA or RNA can allow for a comprehensive genomic and transcriptomic analysis to detect heterogeneous and shared mutational signatures within and between different cancer types [171,172]. This could provide better insight into a patient’s genetic makeup, which in turn can guide clinicians for better disease management. The challenges in sequencing the CTC genome and transcriptome include low abundance in the bloodstream, making their isolation difficult [173], significant genetic heterogeneity compared to the primary tumor, which can affect the capture of the full spectrum of mutations present [15], and the need for sophisticated bioinformatics tools and computational resources for data analysis and interpretation [174]. However, improvements in the isolation and capture of CTCs, the ease of use of different analytical tools, and the removal of biases introduced during sample and sequencing library preparation are increasing the possibility of detecting rare mutations in CTCs [175,176]. By targeting mutational hotspots, a combination of whole-genome amplification, PCR, and Sanger sequencing was used to examine point mutations in the KRAS, BRAF, and PIK3CA genes in CTCs of patients with CRC. The authors detected single-cell mutations in the genes. Interestingly, these mutations were not found in the corresponding tumors [177]. Gkountela and colleagues conducted a comparative analysis of the methylation landscape of single CTCs and CTC clusters from breast cancer patients and mouse models to reveal similarity patterns to embryonic stem cells and identified pharmacological agents that can target clustering, blunt metastasis spreading, and suppress stemness [61]. They observed hypomethylation in the binding sites for OCT4, NANOG, SOX2, and SIN3A of CTC clusters, similar to embryonic stem cells, and identified an FDA-approved Na^+^/P^+^ ATPase inhibitor that enabled the dissociation of CTC clusters into single cells. Thus, they established a link between specific DNA methylation changes and the promotion of cancer stemness and metastasis and pointed to cluster-targeting compounds to suppress the spread of cancer [61]. In a study investigating somatic mutations present in CTCs from colorectal cancer patients, the frequencies of somatic mutations present in the CTCs correlated with prognostic markers and resembled mutational signatures present in the primary tumor [178]. These findings have resulted from significant improvements in the isolation and enrichment of CTCs and efficient whole-genome amplification (WGA) and NGS workflows that reduce bias with improving coverage, sensitivity, and specificity [179,180]. As more targeted cancer panels become increasingly available, the possibility of detecting more actionable mutations will increase, with lower costs and minimal requirements for specialized equipment [181].

### 5.4. In Situ Analysis

Techniques that enable in situ analysis of CTCs directly within the bloodstream or in patient-derived samples provide a real-time understanding of their behavior. The advantage of in situ analysis is that challenges with traditional CTC isolation methods, which compromise CTC integrity, are avoided. These approaches also eliminate potential biases introduced by ex vivo isolation and culture, offering a more accurate representation of CTC biology. CellCollector CANCER01 (DC01) is an approach developed for the in vivo isolation of CTCs from the peripheral blood of cancer patients [182]. The DC01 device enables in vivo isolation and enumeration of CTCs by capturing EpCAM-positive CTCs through a detector placed in the peripheral vein of cancer patients via a cannula [183]. This device has been shown to be tolerated for 30 min in vivo by patients without any side effects [184]. In a study evaluating the feasibility of CTC detection and monitoring in high-risk non-metastatic prostate cancer (PCa) patients undergoing radiotherapy, the authors show that CTCs could be detected before and after radiotherapy using the DC01 device in vivo [184]. Figure 2a below shows images of the captured CTCs. The CTCs were positive for pan-cytokeratin (pan-CK; green) and negative for CD45-red (Figure 2a). This study used the human prostate cancer cell line LNCAP as a positive control (Figure 2b), with non-specific binding lymphocytes as an internal negative control (Figure 2b,c). Three CTC clusters, each containing 2–3 CTCs, were detected in samples from two patients, with total CTC counts of 10 and 15, respectively (Figure 2d).

In the medical community, simple, easy, and sensitive methods to monitor tumor growth are essential for patient care [185]. These may come in the form of improvements or modifications to existing technology, allowing for economical and effective characterization of CTCs. For instance, confocal line scanning, which can be implemented in any confocal microscope system, helped detect CTCs and CTC clusters in vivo in the blood vessels of live mice [186]. Kuo and colleagues also used multi-photon microscopy and antibody-conjugated quantum dots to image CTCs in real-time in living animals [60]. They detected different subpopulations of CTCs, especially those with high metastatic potential—CD24^+^ and CD133^+^ CTCs [60]. These simple and direct noninvasive strategies are needed to detect and investigate the mechanistic underpinnings of tumor metastasis and functionally characterize CTCs.

### 5.5. Functional Assays

Ex vivo culture of CTCs and CTC-derived animal models are gaining increasing popularity as tools to study functional properties, such as drug response and metastatic potential [14,187,188]. These models provide insights into CTC behavior beyond molecular characterization [102]. Additionally, functional characterization of CTCs provides insights into potential treatment options [188]. However, there is a lack of standardized protocols for culturing CTCs of different cancer cell types, and optimal conditions are undefined [189,190]. Current culture conditions involve a cocktail of growth factors, like epidermal growth factor (EGF) and fibroblast growth factors (FGF2 and FGF10), cytokines, hormones, tissue organ extracts, and insulin/insulin-like growth factor 1 (IGF-1) [191,192]. In a recent study optimizing cell culture conditions using a model of primary ovarian cells, Cobalt(II) chloride (CoCl_2_) was used as a hypoxia-mimicking agent [14]. A CoCl_2_ concentration between 100 and 150 μM in culture media was shown to lead to a significant increase in cell counts, demonstrative of a sustainable proliferative activity. However, when the concentration of CoCl_2_ was increased to 200 μM, there was a significant cell cycle arrest at G0/G1 [14]. Efforts like this highlight the importance of refining cell culture conditions to ensure optimal cell growth. Regardless, once successful cultures have been established, the cells can be characterized to provide insight into their biology. Figure 3 shows a phenotypic characterization of CTCs after ~2 weeks in culture [187]. Expression of epithelial markers was observed in 18.18% of samples (number of samples, *n* = 22), while 47% were vimentin-positive (*n* = 17) [187]. These findings corroborate existing knowledge on the loss of epithelial features and the gain of mesenchymal properties following cell culture [193,194,195].

Expansion of CTCs ex vivo via cell cultures also provides avenues for drug sensitivity testing and allows for functional assays like proliferation, migration, invasion, and apoptosis to be performed. In a study performing drug sensitivity profiling of cultured CTCs (treatment-naïve) from patients with SCLC, the authors reported concordance between the response to standard-care chemotherapy (cisplatin and etoposide) administered to the patients and the sensitivities of corresponding cultured CTCs treated with cisplatin and etoposide [188]. Following treatment, response to chemotherapy can also assessed. CTCs can be evaluated for features of viability, early apoptosis, and late-stage apoptosis/necrosis [196]. In breast cancer, CTCs with features of early apoptosis are prognostic of metastasis-free survival and correlate with neoadjuvant chemotherapy response, while the presence of late-stage apoptotic CTCs is associated with a poor response to neoadjuvant chemotherapy and metastasis-free survival [196]. Such findings support the feasibility of predicting treatment outcomes in patients using CTCs. Table 3 summarizes the different methods for characterizing CTC biology, their advantages, and disadvantages.

## 6. Future Perspectives

CTCs are present in very low numbers compared to other blood cells, making their isolation akin to finding a needle in a haystack. Therefore, advanced technologies with high sensitivity are required to capture and detect these rare cells efficiently. Real-time monitoring of CTC dynamics, capturing changes in CTC numbers and characteristics over time, is crucial for understanding the temporal aspects of cancer progression, treatment response, and the emergence of resistance. The development of new high-throughput technologies allows for the rapid and automated processing of large volumes of blood, enhancing the efficiency of CTC isolation. This is particularly important for clinical applications where timely results are essential. Integrating CTC research findings into routine clinical practice, however, requires standardization, validation, and collaboration between researchers and healthcare professionals. These initiatives should be focused on the validation of CTC-based assays, ensuring their reliability and reproducibility. Additionally, making CTC analysis clinically relevant requires the adaptability of existing or new technologies to routine clinical practice. Accordingly, user-friendly platforms need to be integrated with existing diagnostic workflows and point-of-care systems. The integration of advanced analytical platforms, such as single-cell sequencing and next-generation sequencing, will enable comprehensive profiling of CTCs and increase accessibility while significantly decreasing time and associated costs. Thus, the use of CTCs in cancer diagnosis, prognosis, and monitoring treatment outcomes has the potential to revolutionize our approach to cancer detection and treatment.

## 7. Conclusions

Recent advances in CTC research have deepened our understanding of their multifaceted roles in cancer progression, metastasis, and treatment resistance, contributing to the development of more effective diagnostic and therapeutic strategies. These breakthroughs are translating into practical applications with direct benefits for cancer patients. As these findings become more integrated into routine clinical practice, the promise of CTC-based approaches lies in improving cancer diagnosis, prognosis, and treatment outcomes for individuals across diverse cancer types.

## Figures and Tables

**Figure 1 cancers-16-01213-f001:**
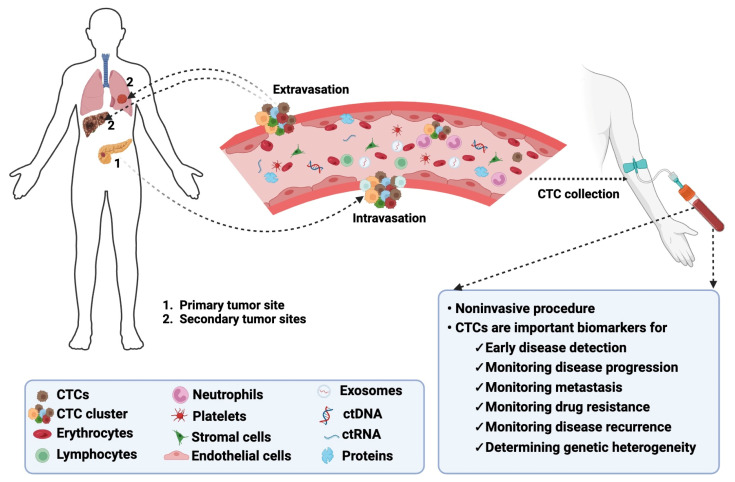
Biology of CTCs in cancer. CTCs detach from their primary tumors, intravasate into the bloodstream, and extravasate from the bloodstream to colonize secondary tumors or metastatic sites. CTCs can be detected following noninvasive collection procedures and serve as biomarkers for monitoring multiple disease aspects. The image was created using BioRender.com.

**Figure 2 cancers-16-01213-f002:**
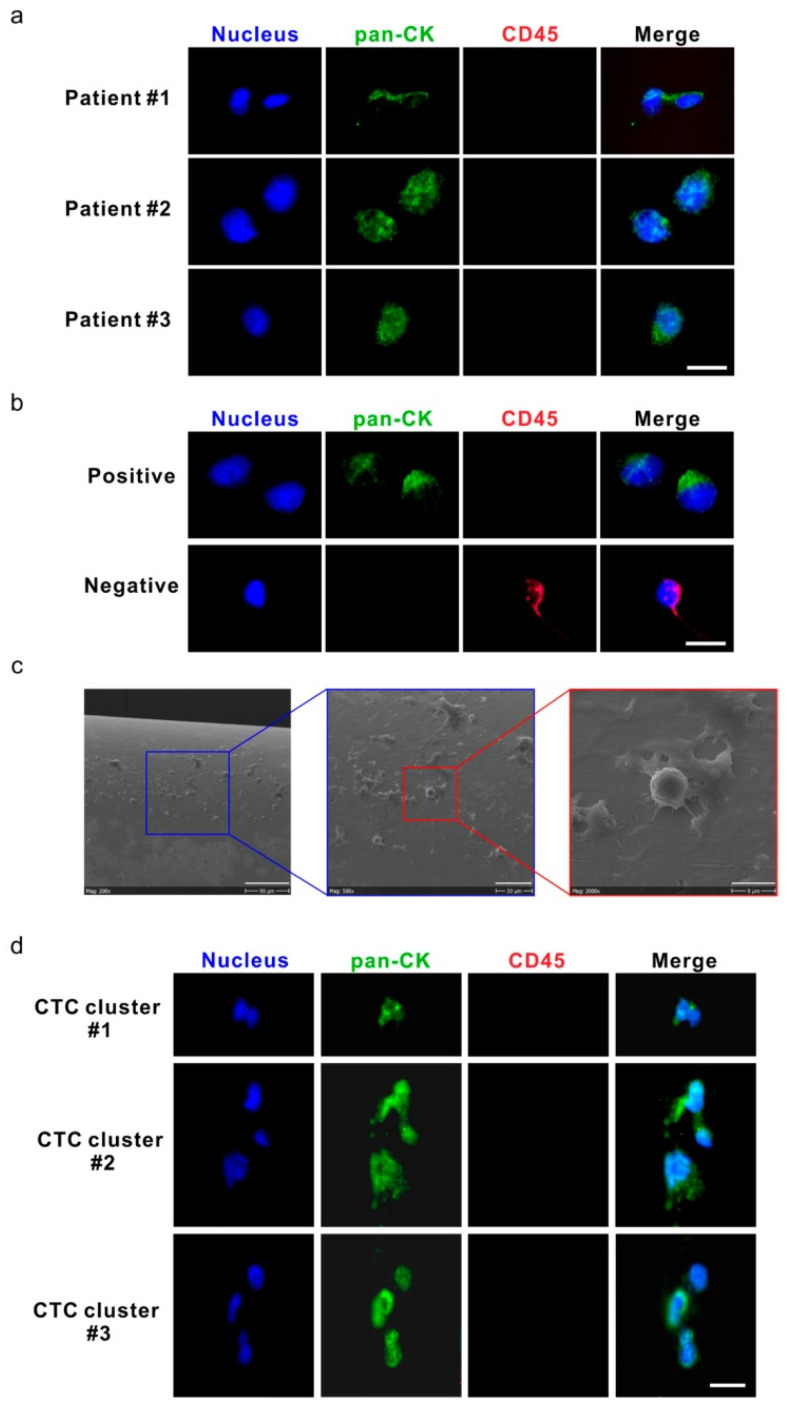
Circulating tumor cells (CTCs) and CTC clusters were detected by the DC01 device in patients with high-risk non-metastatic prostate cancer (PCa). (**a**) Micrographs of five CTCs on DC01 detected from three patients; CTCs were defined as being positive for nucleic staining and pan-CK (pan-cytokeratin) and negative for CD45 by immunofluorescence staining (DNA (blue) and pan-CK (green)). Scale bar: 20 μm. (**b**) Images of control samples for CTC staining panel, upper panel: LNCaP cells captured on DC01 as positive controls for the CTC staining panel by immunofluorescence staining (DNA (blue) and pan-CK (green)); lower panel: leukocytes attached to DC01 served as negative controls for the CTC staining panel by immunofluorescence staining (DNA (blue) and CD45 (red)). Scale bar: 20 μm. (**c**) Scanning electron microscope image showing the surface of DC01 and blood components, including unspecific binding of leucocytes onto DC01. Scale bar (from left to right): 90 μm, 30 μm, and 9 μm. (**d**) Three CTC clusters on DC01 detected from two patients; CTC clusters were found as a cluster of CTCs being positive for nucleic staining and pan-CK and negative for CD45 by immunofluorescence staining (DNA (blue) and pan-CK (green)). All CTCs were negative for PSA, so this channel is not shown. Exposure time for each channel: nucleus DNA, 800~1000 ms; pan-CK, 3000 ms; CD45, 6000 ms. Scale bar: 20 μm. Source: Reprinted from [184] under an open-access Creative Commons CC BY 4.0 license.

**Figure 3 cancers-16-01213-f003:**
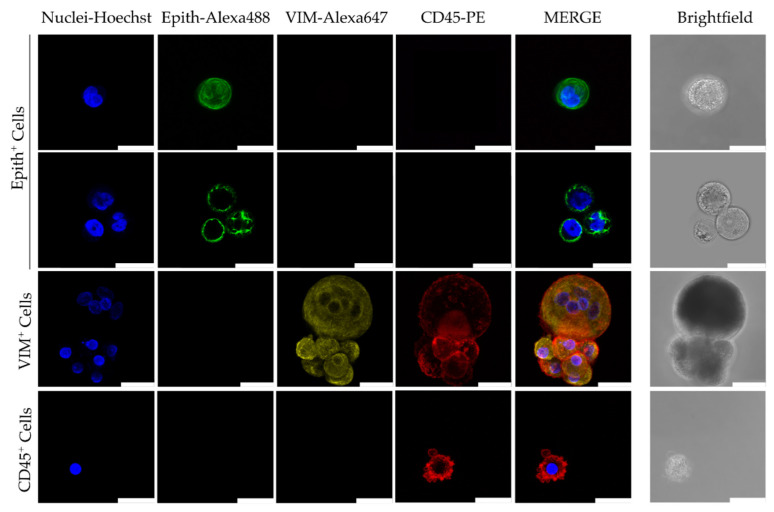
Representative images by confocal microscopy of immunofluorescence characterization of circulating tumor cells (CTCs) after culture. Immunofluorescent staining was performed using a combination of anti-human Epithelial markers (Epith: EpCAM, E-Cadh, and PanCK) (in green), anti-Vimentin (VM, in yellow), and anti-CD45 (in red). Scale bar represents 25 μm. Source: [187] under an open-access Creative Commons CC BY 4.0 license.

**Table 1 cancers-16-01213-t001:** Challenges associated with detection, isolation, and characterization of CTCs.

Challenges	Description/Effects	Ref.
Low Frequency in Bloodstream	- Extremely rare compared to other circulating cells (e.g., blood cells) - Ranges from a few to a few hundred CTCs per milliliter of blood - Low frequency makes detection and isolation difficult	[93,94]
Heterogeneity	- Exhibits both genetic and phenotypic variabilities - Reflects cellular diversity within the primary tumor - Heterogeneity complicates efforts to capture a representative sample of CTCs for analysis	[95,96,97]
Cell Viability	- CTCs are fragile and can be damaged during isolation processes - Isolation and analysis of viable CTCs are crucial for meaningful downstream studies - Lack of viable cells may affect subsequent functional assays	[98]
Dynamic Changes in CTC Numbers	- Number of CTCs in the bloodstream vary over time - Number of CTCs is affected by tumor size, treatment effects, and stage of the disease - Variable CTC numbers add complexity to studying CTCs longitudinally	[14,99]
Contamination from Normal Cells	- Isolation of CTCs can be complicated by contamination from normal blood cells, such as leukocytes - Presence of other cells reduces the purity of CTC samples - Contaminating cells can interfere with downstream analyses	[100]
Technical Limitations	- Traditional methods for CTC isolation, such as density gradient centrifugation, do not efficiently capture CTCs due to their similar size and density compared to other blood cells - Newer technologies, like microfluidic devices and immunomagnetic separation, need refinements to achieve high purity and recovery rates	[101,102]
Lack of Standardization	- Lack of standardized protocols for CTC isolation and characterization - Different isolation methods and technologies may yield varying results - Lack of standardization makes it challenging to compare data across studies, leading to potential discrepancies in the interpretation of findings	[26,103,104]
Ethical and Consent Issues	- Obtaining blood samples for CTC analysis requires informed consent - Collecting longitudinal samples to monitor disease progression may have psychological impacts on patients	[105]
Clinical Relevance	- Establishing the clinical relevance of CTCs and their role as prognostic or predictive biomarkers requires large-scale clinical validation - Research findings are not integrated with routine clinical applications	[106,107]
Cost and Accessibility	- Advanced technologies for CTC isolation and analysis are expensive - The cost associated with isolation and analysis methods limit their widespread implementation and use - Advanced technologies needed are not readily accessible in all healthcare settings	[108,109]

**Table 2 cancers-16-01213-t002:** CTC isolation methods—advantages and disadvantages.

Method	Principle	Pros	Cons	Refs.
Density Gradient Centrifugation	Differential centrifugation separates blood components based on their density, allowing for the isolation of CTCs	Simple, cost-effective	- Limited specificity - Contamination from normal blood cells	[129]
Filtration Techniques	Filters with defined pore sizes are used to physically separate CTCs from blood cells based on size	Simple, cost-effective	- Risk of clogging - Loss of smaller CTCs	[130,131,132]
Epithelial Cell Adhesion Molecule (EpCAM) Enrichment	EpCAM, a cell surface marker often expressed in epithelial cancers, is targeted for CTC enrichment	Commonly used, FDA-approved platforms	- EpCAM-negative CTCs may be missed - Loss of CTC heterogeneity	[120]
Immunomagnetic Separation	Antibodies specific to tumor- associated antigens are used to capture CTCs by attaching to magnetic beads	High specificity, potential for enrichment of viable CTCs	- Limited by the availability of specific antigens - Loss of CTC viability	[133,134]
Microfluidic Devices	Microscale devices use various mechanisms, such as size-based filtration or antibody-coated surfaces, to isolate CTCs from blood	High throughput, potential for single-cell analysis, and minimal sample processing	- Requires efficient capture to avoid CTC damage - Requires standardization of devices	[125,135]
Fluorescent- Activated Cell Sorting (FACS)	Fluorescently tagged cells are separated by flow cytometry	High specificity and throughput for CTC enrichment	- Requires a high number of input cells	[136,137]

Notes: Density gradient centrifugation and filtration techniques are label-independent. EpCAM enrichment, immunomagnetic separation, and FACS are label-dependent. Microfluidic devices can be either label-dependent or label-independent.

**Table 3 cancers-16-01213-t003:** Summary of methods used in characterizing CTC biology.

Method	Advantages	Disadvantages
Polymerase chain reaction	- Detect specific DNA sequences - High sensitivity and specificity	- Does not provide any information about genetic heterogeneity - Improvements needed to detect rare mutations
Single-cell analysis	- Allows for the analysis of individual CTCs to understand heterogeneity - Provides insights into genetic and phenotypic diversity	- Requires specialized techniques and analysis tools - Analysis can be time-consuming
Next-generation sequencing	- Allows for high-throughput sequencing of CTC genomes - Provides comprehensive genomic information and detects rare mutations	- Expensive, complex data analysis - May require high DNA input
In situ analysis	- Examines CTCs directly in the blood - Offers spatial information and preserves CTC integrity	- Requires specialized imaging techniques - Not suitable for high-throughput analysis
Functional assays	- Evaluates CTCs’ functional properties, like proliferation, migration, and drug response - Involves ex vivo culture of CTCs	- Assay conditions may not fully represent in vivo conditions - Lack of standardized culture protocols - Variable results

## Data Availability

The data can be shared upon request.

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
