# Peer review of "Advancements in Circulating Tumor Cell Research: Bridging Biology and Clinical Applications"

_cancers, 2024, doi:10.3390/cancers16061213_

Round 1

Reviewer 1 Report

Comments and Suggestions for Authors

This review discusses recent findings about the detection and characterization of CTC. The authors claim that integrating these findings into routine clinical workflows promises to revolutionize the diagnosis and treatment of cancer.Some points should be noticed as below,

1) “Abstract: Circulating tumor cells (CTCs) are cells released from the primary tumor and intravasate into the blood or lymphatic vessels”, it might have other situations, such as tumor cells that have metastasized in distant organs spreading back into circulation (the blood or lymphatic vessels). If there is, such a description in this manuscript needs to be modified accordingly.

2) Continuing with the above questions, setting aside the clinical effects of CTC on patient prognosis and treatment, what do you think are the main biological functions of CTC in cancer? It is well known as precursors of metastasis? In turn, whether it has a promoting effect on the primary tumor?

3) What is the interaction between CTC and peripheral blood cells such as monocytes and neutrophils during the circulatory process? It is recommended to talk briefly.

4) “2.4. Metastasis and Disease Progression...Metabolic reprogramming of CTCs enables them to survive...,  one recent paper unprecedentedly proposes a "Anti-Warburg Effect" (AWE) in CTCs-a metabolic shift bridging primary tumors and metastases (https://pubmed.ncbi.nlm.nih.gov/38300633/). It should further help for this issue. 

Reviewer 2 Report

Comments and Suggestions for Authors

The review paper by Salu, et al. provides a comprehensive overview of the CTC research and application. The paper is well written and very clear. I have only a couple minor suggestions:

1.        Table 2 summarizes the CTC isolation methods. It might be helpful to also indicate which ones are label-dependent and label-independent.

2.        A table comparing and summarizing the different methods for CTC analysis (e.g. PCR, single-cell, NGS and in situ analysis) may improve the clarity of the paper.

Round 2

Reviewer 1 Report

Comments and Suggestions for Authors

This review fully meets the requirements for publication now.